# Pretreatment, Anaerobic Codigestion, or Both? Which Is More Suitable for the Enhancement of Methane Production from Agricultural Waste?

**DOI:** 10.3390/molecules26144175

**Published:** 2021-07-09

**Authors:** Lütfiye Dumlu, Asli Seyhan Ciggin, Stefan Ručman, N. Altınay Perendeci

**Affiliations:** 1Department of Environmental Engineering, Akdeniz University, Antalya 07070, Turkey; lutfiye.dumlu@csb.gov.tr (L.D.); asliciggin@akdeniz.edu.tr (A.S.C.); 2Ministry of Environment and Urbanization, General Directorate of Environmental Management, Department of Water and Soil Management, Ankara 06510, Turkey; 3Analytical Science and National Doping Test Institute (ASNDTI), Mahidol University, 272 Rama 6 Road, Bangkok 10400, Thailand; stefan.ruc@mahidol.ac.th

**Keywords:** agricultural wastes, anaerobic codigestion, hydrolysis rate, first-order kinetic, lignocellulosic residues, thermochemical pretreatment

## Abstract

Pretreatment and codigestion are proven to be effective strategies for the enhancement of the anaerobic digestion of lignocellulosic residues. The purpose of this study is to evaluate the effects of pretreatment and codigestion on methane production and the hydrolysis rate in the anaerobic digestion of agricultural wastes (AWs). Thermal and different thermochemical pretreatments were applied on AWs. Sewage sludge (SS) was selected as a cosubstrate. Biochemical methane potential tests were performed by mixing SS with raw and pretreated AWs at different mixing ratios. Hydrolysis rates were estimated by the best fit obtained with the first-order kinetic model. As a result of the experimental and kinetic studies, the best strategy was determined to be thermochemical pretreatment with sodium hydroxide (NaOH). This strategy resulted in a maximum enhancement in the anaerobic digestion of AWs, a 56% increase in methane production, an 81.90% increase in the hydrolysis rate and a 79.63% decrease in the technical digestion time compared to raw AWs. On the other hand, anaerobic codigestion (AcoD) with SS was determined to be ineffective when it came to the enhancement of methane production and the hydrolysis rate. The most suitable mixing ratio was determined to be 80:20 (Aws/SS) for the AcoD of the studied AWs with SS in order to obtain the highest possible methane production without any antagonistic effect.

## 1. Introduction

As a result of growing energy demands and the rising importance of sustainable waste management, the number of studies evaluating different types of wastes for energy production has increased. It is well known that agricultural wastes produced in high quantities are a significant feedstock containing lignocellulosic residues for second-generation biofuel production [1]. On a local scale, agricultural wastes (AWs) consisting of roots, stalks, leaves and fruits originating from the production of tomato, pepper, cucumber, eggplant and courgette are the main causes of environmental problems in Antalya, Turkey [2]. Conventional disposal methods for these wastes, such as landfilling, uncontrolled burning and unconfined storage in the territory, result in significant environmental issues in urban areas [3].

Anaerobic digestion (AD) is described as an energy-efficient biotechnological process that combines sustainable waste disposal and renewable energy production. AD is frequently studied for the conversion of agricultural wastes to biogas and biofertilizer [4]. The lignocellulosic nature and high carbon-to-nitrogen (C/N) ratio of agricultural wastes are considerable obstacles that significantly affect biogas production efficiency [5,6]. The main components of the lignocellulosic residues are cellulose, hemicellulose and lignin. Although cellulose and hemicellulose (holocellulose) are valuable carbohydrate components for biogas production, lignin is the main barrier in biogas production from holocellulose [7,8]. Pretreatment is a useful method for increasing the accessibility of holocellulose by breaking down the physical lignin barrier. In addition to reducing lignin content, pretreatment increases the biodegradability of holocellulose [8,9].

Physical, chemical, thermal and biological processes can be applied for pretreatment of lignocellulosic residues. The most effective pretreatment conditions should be determined for each lignocellulosic residue to increase the biodegradability of biomass and the acceleration of biogas production, as well as the optimization of cost and energy [8,9]. Thermal pretreatment is the simplest and cheapest method when it is applied as a low-temperature heat application. Thermal pretreatment can increase biogas production while decreasing the duration of the digestion time [10]. One of the drawbacks of this process is the possible occurrence of Maillard reaction when thermal pretreatment is applied at a high temperature or at a low temperature with long reaction times [11]. In order to keep the reaction time short and increase methane production, it has been proposed to combine low-temperature thermal pretreatment with chemical pretreatment [11].

Chemical pretreatment is also widely applied for the enhancement of anaerobic digestion by disintegrating the lignocellulosic structure. The main chemicals used in chemical pretreatment are acidic agents such as sulfuric acid and hydrochloric acid [2,3], alkaline agents such as sodium hydroxide and calcium hydroxide [2,12] and oxidants such as hydrogen peroxide [1]. While acidic pretreatment facilitates the hydrolysis of hemicellulose and the interruption of lignin [8], alkaline pretreatment provides the depolymerization of lignin and an increase in the porosity and surface area through solvation and saponification reactions [9,13]. The incorporation of thermal pretreatment into chemical pretreatment has been a subject of several studies due to its benefits such as decreased chemical consumption and increased biogas production [1,2,3,4,5,6,14,15,16]. Although thermochemical pretreatment with alkaline agents has been recommended for agricultural wastes with low lignin content, the most efficient chemical agent should be determined for each waste/feedstock with a different lignocellulosic structure [13].

In addition to pretreatment, anaerobic codigestion (AcoD) has been proven to be an efficient strategy for the enhancement of biogas production. AcoD is the anaerobic digestion of at least two different types of organic waste [17]. In addition to an increase in biogas production due to the positive synergistic effect of different biomasses with complementary characteristics (especially in terms of the C/N ratio), AcoD has numerous advantages such as the utilization of larger centralized bioreactors, the supply of missing nutrients through the cosubstrate and the treatment of toxic substances through cometabolism [18,19]. In the AcoD, it is very important to determine the most suitable cosubstrates and mixing ratio to maximize methane production with a synergistic effect and prevent the production of inhibitory components that can cause an antagonistic effect [17,19]. Sewage sludge, known to have a high nitrogen content, has been preferred as a cosubstrate to provide the optimum C/N ratio in the anaerobic digestion of agricultural wastes with a high carbon content [20,21,22]. Codigestion of agricultural wastes with sewage sludge (SS) can increase biogas production by (i) maintaining an optimal pH for bacteria, (ii) decreasing free ammonia/ammonium inhibition and (iii) providing a better C/N ratio during anaerobic digestion [18]. Although the synergetic effect of the AcoD of wheat straw and corn stalk with SS has been established [20,23], there is limited information regarding the AcoD of other agricultural wastes with SS.

A batch anaerobic biodegradability test, namely the biochemical methane production (BMP) test, is a useful tool to determine the synergetic and antagonistic effects on the AcoD of cosubstrates and the optimum mixing ratio of cosubstrates [24]. Additionally, the BMP test provides valuable information on the methane production potential, the technical digestion time and the hydrolysis kinetics [18]. The technical digestion time is defined as the time (of day) at which 80% of the maximum methane production is completed [25]. The technical digestion time obtained with the BMP test has been recommended to be used as an indicator when determining the hydraulic retention time in continuous systems [26]. The hydrolysis rate is another important parameter for determining the effect of a process such as pretreatment and AcoD on anaerobic digestion. Generally, any improvement in the rate of hydrolysis makes the process economically more attractive, given that hydrolysis is usually assumed to be a rate-limiting step in anaerobic digestion. First-order kinetics are generally used for the determination of the hydrolysis rate based on the results of the BMP test [1,5].

Because of the numerous benefits of the pretreatment and AcoD on anaerobic digestion, as summarized above, their integration has the potential to increase the performance of anaerobic digestion. In order to maximize methane production, it is important to determine a suitable pretreatment method depending on the type of the substrate and cosubstrates [27]. In this context, this study focused on the enhancement of methane production from agricultural wastes by evaluating the effect of thermal and different thermochemical pretreatment methods on the monodigestion and AcoD of agricultural wastes (AWs) with sewage sludge (SS). For this purpose, batch BMP tests were carried out by mixing raw or pretreated AWs with SS at different ratios. Thermochemical pretreatment was applied to AWs under alkaline conditions with sodium hydroxide (NaOH) and calcium hydroxide (Ca(OH)_2_) and under acidic conditions with sulfuric acid (H_2_SO_4_) and hydrochloric acid (HCl). In addition to the cumulative methane production, the effects of pretreatment and AcoD on the anaerobic digestion process were evaluated by comparing the technical digestion times and hydrolysis rates.

## 2. Results and Discussion

### 2.1. Effect of Thermal and Thermochemical Pretreatment on Cumulative Methane Production (CMP) from AWs

The CMP, along with the chemical oxygen demand (COD) and volatile solid (VS) removal efficiencies obtained from raw AWs and pretreated AWs after 100 days of anaerobic digestion, is presented in Figure 1. As seen in Figure 1a, while thermal pretreatment (AW1) and thermochemical pretreatment with alkaline agents (AW4 and AW5) provided an increase in the COD removal efficiencies, thermochemical pretreatment with acidic agents (AW2 and AW3) caused a decrease in COD removal. This observation indicates that thermal pretreatment and thermochemical pretreatment with alkaline agents were effective in the mineralization of organic matter.

On the other hand, the VS removal efficiency increased in all pretreated AWs compared to raw AWs. While the highest COD removal rate was achieved by thermochemical pretreatment with NaOH (AW4), thermal pretreatment (AW1) resulted in the highest VS removal rate. In a study investigating the effects of thermal and different thermochemical pretreatment methods on food waste, it was also observed that thermal pretreatment resulted in a more efficient VS removal than thermochemical pretreatment with NaOH [28]. It was stated that a lower VS removal rate was caused by the role of NaOH in removing total solids (TSs), rather than VS, due to total solid mineralization with NaOH pretreatment [28].

The anaerobic digestion of raw AWs provided 284.27 mL/gVS of CMP and 60.49% COD and 26.80% VS removal efficiencies. The cellulose, hemicellulose and lignin composition of AWs found in this study is similar to that reported separately for cucumber, eggplant and tomato crop residues in Reference [16]. However, the highest CMP of 124.4 mL/gVS was reported for the anaerobic digestion of tomato crop residues by Li et al. [16]. The reason for the high BMP value obtained in this study may be due to the use of various residues, with a predominance of tomatoes (61.71%), compared to the result of Li et al. [16].

Thermal pretreatment without any chemicals resulted in a slight increase in the CMP (15%) compared to raw AWs. Similarly, thermal treatment at 121 °C for 60 min was reported to provide an increase of 29% and 11% in the CMP from wheat straw and sugarcane bagasse, respectively [1]. It should be noted that because of the limited number of studies with similar agricultural wastes, the results were predominantly discussed by comparing them with intensively researched wastes such as wheat straw, fruit and vegetable wastes and food wastes.

When AWs were subjected to thermochemical pretreatment with NaOH (AW4). The highest CMP was obtained as 443.53 mL, which was 56% higher than the CMP of raw AWs. Similarly, in a recent study investigating the effect of thermochemical pretreatment with acids (H_2_SO_4_, HCl and phosphoric acid (H_3_PO_4_)) and a base (NaOH) on the CMP of grass lawn, the highest increase in the CMP was 25.7% when thermochemical pretreatment with 20% NaOH was applied at 80 °C [29]. The positive effect of thermochemical pretreatment with NaOH on the CMP from different lignocellulosic wastes was also reported. The increase in the CMP was reported as 32–67% compared to untreated control when the thermochemical pretreatment was applied to wheat straw with NaOH [1,14]. Thermochemical pretreatment with 2.5% NaOH at 100 °C resulted in a 23% increase in the CMP compared to raw grass silage [18]. Additionally, a 37% increase was observed in the CMP compared to raw sunflower residues after thermochemical pretreatment with 4% NaOH at 55 °C [6].

As illustrated in Figure 1b, thermochemical pretreatment with H_2_SO_4_ (AW2) resulted in a decrease in the CMP compared to raw AWs. Similarly, a decrease was observed in the CMP with pretreatment using 2% H_2_SO_4_, while pretreatment with 2% NaOH increased the CMP by 94% compared to raw wheat straw [13]. Although it was proven that introducing high acid and alkaline concentrations for pretreatment results in the production of inhibitory substances [2], it was not expected for the CMP to be inhibited at very diluted acid concentration (0.1% H_2_SO_4_) used in this study. As with pretreatment with H_2_SO_4_, inhibition in biogas production was observed when thermochemical pretreatment with 1.5% HCl was applied at 121 °C for 60 min to wheat straw and sugarcane bagasse [1]. The inhibition of biogas production was confirmed by the detection of hydroxymethylfurfural (HMF) and furfural in the samples after treatment. A similar inhibition effect was also observed in the thermochemical pretreatment of the fruit and vegetable wastes (FVWs) with 2.5% HCl [2]. While inhibition was observed in the above-mentioned studies, thermochemical pretreatment with 0.1% HCl resulted in a 29.97% increase in the CMP compared to raw AWs in this study (Figure 1b). Because this difference may be attributed to applied HCl concentrations, optimization should be performed when determining the levels of HCl in thermochemical pretreatment to prevent the inhibition of biogas production.

Thermochemical pretreatment with Ca(OH)_2_ (AW5) also resulted in the production of 37.08% more methane compared to raw AW (Figure 1b). Similarly, it has been reported that pretreatment of rice straw with 0.5% and 2% Ca(OH)_2_ at 80 °C resulted in an increase of 25.73% and 34.75%, respectively, in the CMP compared to untreated control [12]. In another study, a 20.38% increase in the CMP was obtained from wheat straw when wheat straw was pretreated with 0.59% Ca(OH)_2_ for 48 h at 20 °C [15]. In summary, thermochemical pretreatment with alkaline agents was determined as the most effective pretreatment method in methane production from all studied AWs. When deciding on the chemical substance to be used for pretreatment, it may be possible to prefer cheaper but slightly less efficient Ca(OH)_2_ instead of NaOH in cases where optimization of anaerobic digestion cost is a priority [30].

### 2.2. Effect of AcoD with SS on CMP from AWs

The first sets of AcoD studies were performed to evaluate the AcoD of raw AWs with SS. For this purpose, BMP tests were carried out for the AcoD of raw AWs with SS at mixing ratios of 20, 40, 50, 60 and 80 and compared with the monodigestion of pretreated AWs and untreated SS. The COD and VS removal efficiencies along with the CMPs obtained from the AcoD of raw AWs and SS are presented in Figure 2.

In addition to the lowest COD and VS removal efficiencies, the lowest CMP was observed in the digestion of SS alone as 176.67 mL/gVS. The measured CMP was 187.42 mL/gVS in a study evaluating the effect of pretreatment on the AcoD of rice straw with waste-activated sludge [20]. In another study on the AcoD of FVW with the SS, the measured CMP was 119 mL/gVS in the monodigestion of primary sludge [21]. Similarly, the CMPs were determined to be 133 and 190 mL/gVS for SS consisting of primary sludge and waste-activated sludge in studies performed to determine a suitable mixing ratio for the AcoD of SS with food waste [22,31]. The values for the CMP of SS reported in the above-mentioned studies are quite similar to the CMPs obtained in the present study.

Considering AcoD, the highest CMP of 266.85 mL/gVS was obtained from the sample of raw AWs with SS at a mixing ratio of 80:20 (AWs/SS) (Figure 2b). In the study carried out for the AcoD of FVW with primary sludge (PS), the highest observed CMP was 89.8 mL/gVS at a mixing ratio of 50:50 (FVW/PS) by Lahoz et al. [32]. In the AcoD of FVW generated in a wholesale market with SS, the optimum mixing ratios were determined to be 60:40 and 80:20 (FVW/SS) [33]. Additionally, in a mixture similar to the one used in our study, an increment in the CMP was observed with the increase in the volumetric percentage of FVW. On the other hand, in a recent study focused on the AcoD of SS with FVW consisting of cabbage, eggplant, zucchini, potato, broccoli, tomato and nectarine, the highest CMP (127 mL/gVS) was achieved when the ratio of sludge in the mixture was high (25:75 (FVW/SS)) [34]. Although this study focused on methane production and the hydrolysis rate of agricultural wastes and sewage sludge by anaerobic codigestion, it should be noted that different kinds of waste materials such as brewery spent grain, palm oil mill, spent coconut cobra, manure and cow urine as a cosubstrate [35,36] have been codigested from the point of methane yields and improved biofertilizer properties of digestate.

The interaction between substrates used in AcoD can result in synergistic or antagonistic effects. These effects can be determined by comparing the measured CMPs with the theoretical methane production [4,20,37]. The theoretical CMP has been calculated by multiplying the specific CMPs obtained from the monodigestion of codigested waste by the percentage of that codigested in the AcoD [10,20,37]. In line with this information, the theoretical CMPs and synergistic and antagonistic constant (α) were calculated and are presented in Table 1. α is the rate representing the synergistic and antagonistic effect on AcoD. Unfortunately, there was no significant antagonistic or synergistic effect observed in any mixing ratio. This indicates that the AcoD of raw AWs with SS does not cause any synergistic or antagonistic effects similar to that obtained in the AcoD of wheat straw with manure [38] and microalgae with primary sludge [10]. Furthermore, there is no drawback for the usage of AWs and SS from the perspective of waste management and energy production.

### 2.3. Effect of Thermal and Thermochemical Pretreatment on AcoD of AWs with SS

After determining the effect of pretreatment and AcoD on the anaerobic digestion of AWs, these two enhancement strategies were evaluated together in the next step. For this purpose, BMP tests were carried out for the AcoD of pretreated AWs with SS at mixing ratios of 20, 40, 50, 60 and 80 and compared with the monodigestion of pretreated AWs and untreated SS. Thermal pretreatment (AW1) and thermochemical pretreatment using H_2_SO_4_ (AW2), HCl (AW3), NaOH (AW4) and Ca(OH)_2_ (AW5) were implemented to AWs in a similar manner as in Section 2.1. The final COD and VS concentrations were measured in all BMP reactors at the end of 100 days of digestion period in order to evaluate the combined effect of AcoD and pretreatment on COD and VS removal efficiencies (Figure 3).

Compared to monodigestion of AW1, the AcoD of AW1 with SS at a mixing ratio of 80:20 resulted in higher COD and VS removal efficiencies (75.2% COD and 52.4% VS). Additionally, the highest VS removal rate was obtained in the AcoD of AW1 with SS at a mixing ratio of 80:20 compared to all assays. The AcoD of AWs pretreated by acidic agents (H_2_SO_4_ and HCl) provided an increase in the COD removal efficiency compared to the monodigestion of AWs pretreated using acids (AW2 and AW3). On the other hand, the AcoD of AWs pretreated using alkaline agents (NaOH and Ca(OH)_2_) did not result in an increase in the COD and VS removal efficiencies. These findings suggest that thermochemical pretreatment using acidic agents combined with AcoD contributes to the acceleration of COD removal from AWs compared to alkaline agents. Despite the positive effect of acidic agents on COD removal in the digestion of AWs with SS, the highest COD removal efficiency was obtained in the monodigestion of AW4 (pretreated with NaOH).

The average CMPs measured from the AcoD of pretreated AWs with SS are presented in Figure 4.

Figure 4a shows that reducing the ratio of AW1 in the waste mixture below 80% resulted in a significant reduction in the CMP. There were no significant differences observed in the CMPs when AW2 was used in AcoD with SS at different mixing ratios (Figure 4b). This was probably due to the production of inhibitory products when the AWs were pretreated with H_2_SO_4_ similar to what was observed in the monodigestion of AW2 [3].

According to Figure 4c, the CMP obtained from the AcoD of AW3 with SS in the ratio of 80:20 was almost the same as the CMP obtained in the monodigestion of AW3. When the CMP measured from the AcoD of AW3 with SS at a mixing ratio of 80:20 was compared to the theoretical CMP, it was determined that AcoD provided a slightly synergistic effect based on the *α* value of 1.06 for this sample. This is the only sample in this study where AcoD was found to have a synergistic effect on AWs. When AW4 was codigested with SS (Figure 4d), the CMP decreased parallel to the increase in the ratio of SS in the waste mixture.

Although a synergetic effect was observed in the AcoD of rice straw pretreated using NaOH with SS [20] and in the AcoD of wheat straw pretreated using NaOH with SS [39], the AcoD of the studied AWs with SS resulted in no synergetic or antagonistic effect from the point of methane production enhancement of AWs. For the AcoD of AW4 with SS, the optimum mixing ratio was 80:20 with the CMP of 359.44 mL/gVS. Similarly, the highest CMP in the AcoD of AW5 with SS was determined to be 327.13 mL/gVS at a mixing ratio of 80:20 (Figure 4e). In this context, the most appropriate mixing ratio was determined as 80:20 (on VS bases) for the AcoD of the studied AWs with SS considering the waste management in the region.

### 2.4. Evaluation of Anaerobic Digestion Kinetics

Generally, pretreatment improves the process kinetics while enhancing methane production. In this study, kinetic studies were conducted to evaluate the effect of pretreatment on the hydrolysis rate and the technical digestion time of the anaerobic digestion of AWs. It has been suggested that the hydrolysis rate can be determined by evaluating the first days of CMP [24]. Therefore, the hydrolysis rates (*k_h_*) were estimated according to the best fit obtained between the experimental CMP profiles and the model simulations for the first 6 days. The hydrolysis rates obtained from the first-order model that provided a best fit with the CMPs of raw and pretreated AWs are given in Table 2. Model calibration results used for the determination of the *k_h_* values are given in the Appendix A.

The first-order kinetic model provided a good fit for the experimental CMP results with high regression coefficients (*R*^2^) of 0.9689–1.0000. The high correlation coefficients indicated that the first-order kinetic model was able to simulate the CMP profiles correctly. The hydrolysis rates given in Table 2 indicate that pretreatment improved the hydrolysis rate in addition to enhancing methane production. With the application of thermal pretreatment alone, the hydrolysis rate increased by 33.3%. The increases in the hydrolysis rate with thermochemical pretreatment by NaOH, HCl, H_2_SO_4_ and Ca(OH)_2_ were 81.90%, 80.13%, 24.72% and 1.99%, respectively. While high hydrolysis rates were obtained with thermochemical pretreatment using NaOH (AW4) and HCl (AW3), thermochemical pretreatment with Ca(OH)_2_ (AW5) slightly increased the hydrolysis rate. Although thermochemical pretreatment with Ca(OH)_2_ (AW5) provided higher a CMP than thermochemical pretreatment with HCl (AW3), the obtained hydrolysis rates indicate that pretreatment with HCl was more effective in increasing the hydrolysis rate.

In the literature, the first-order kinetic model was usually applied for the determination of the overall rate of anaerobic digestion, and the effect of thermochemical pretreatment on the overall reaction rate (*k_R_*) was evaluated. Therefore, the recorded increases in the reaction rates by pretreatment are referred to here. A 55% increase in the hydrolysis rate was reported when thermochemical pretreatment was applied on sorghum with 10% NaOH at 55 °C [5]. In another study, thermochemical pretreatment of wheat straw with 1% NaOH at 121 °C resulted in a 47.92% increase in the hydrolysis rate [1]. Additionally, a thermochemical pretreatment with 6% NaOH resulted in a 45%, 72% and 88% increase in the hydrolysis rates of giant reed, fiber sorghum and barley straw, respectively [40]. In accordance with the above-mentioned studies, the 81.90% increase obtained in this study supports the finding that thermochemical pretreatment with NaOH is the most suitable treatment method for studied AWs.

Interestingly, although the CMP obtained for AW2 was lower than raw AWs, thermochemical pretreatment with H_2_SO_4_ resulted in an increase in the hydrolysis rate. This inconsistency can be explained by the fact that thermochemical pretreatment with H_2_SO_4_ was effective in increasing hydrolysis of organic material, while on the other hand, methane production was disrupted by inhibitory products such as HMF and furfural. Moreover, the hydrolysis rate obtained for AW2 was higher than that of AW5. It has been reported that the increase in methane production was mostly related to the increase in the hydrolysis rate obtained by pretreatment [41,42]. However, the findings obtained for AW2 show that a high hydrolysis rate cannot always be associated with an increase in methane production. Inhibitors produced during the pretreatment should also be considered for complex microbial populations in anaerobic digestion.

A decrease in the technical digestion time (T_80_) has been identified as an indicator for the acceleration of biodegradation and methane production by solubilization of organic matter with pretreatment [18]. The calculated T_80_ values are given in Table 2. Because of lower methane production in AW2 compared to raw AW, the T_80_ value could not be calculated. The experimental CMP profiles used for the determination of T_80_ values are shown in Appendix A. Although T_80_ was calculated after 54 days for raw AWs, thermochemical pretreatment with NaOH, HCl and Ca(OH)_2_ decreased the T_80_ by 79.63%, 59.26% and 50.00%, respectively. Additionally, a 31.48% decrease was observed in T_80_ after the implementation of the thermal pretreatment alone. The efficacy of thermochemical pretreatment with NaOH in reducing T_80_ was also reported in several studies conducted on lignocellulosic residues. The thermochemical pretreatment of the grass silage with 2.5% NaOH at 100 °C resulted in a 36% decrease in T_80_ [18]. Thermochemical pretreatment with 6% NaOH reduced T_80_ by approximately 50% compared to untreated barley straw [40].

The evaluation of CMPs of AWs pretreated with HCl and Ca(OH)_2_, together with their hydrolysis rates and technical digestion times, showed that these two chemicals had different effects on the anaerobic digestion process. While pretreatment with Ca(OH)_2_ increased methane production, pretreatment with HCl resulted in a faster hydrolysis rate and shorter digestion time. A low hydrolysis rate with a high methane production was also observed with the application of thermochemical pretreatment to wheat straw and sugar cane pulp [1]. In a recent study where pretreatment of *Miscanthus x giganteus* with calcium oxide (CaO) was optimized in terms of methane production and reaction rate, it was also determined that methane production and reaction rate were affected differently by different pretreatment conditions [42].

### 2.5. Evaluation of AcoD Kinetics

Kinetic studies were also performed to evaluate whether AcoD has any effect on the hydrolysis rate of AWs and SS. When the CMPs obtained from the AcoD of pretreated AWs with SS were evaluated, a synergetic effect was observed only in one sample that was exposed to thermochemical pretreatment with HCl (AW3). Therefore, the modified first-order kinetic model was simulated to fit the experimental results gathered from the AcoD of raw AW and AW3 with SS. The model calibration results obtained with best fits in the determination of the *k_h_* values for the AcoD of raw AWs and AW3 with SS are presented in Appendix A, respectively. The *k_h_* values predicted by the first-order model that provided the best fit for the AcoD of AWs and AW3 with SS are given in Table 3.

A modified first-order kinetic model was able to simulate the AcoD kinetics with higher *R*^2^ (>0.973). In the AcoD of raw AWs with SS, the increase in the amount of SS in mixed waste led to a decrease in the hydrolysis rate of AWs, while the kh of SS was increased with the increase in the amount of AWs. This indicates that AcoD with AWs provided a synergistic effect on SS by increasing the hydrolysis rate. On the other hand, the implementation of AcoD with SS had an antagonistic effect on the hydrolysis rate of AWs. A similar trend was also observed in the hydrolysis rate of AW3 which decreased with the addition of SS while the hydrolysis rate of SS was not affected by the presence of the pretreated AWs in mixed waste. A decrease in *k_h_* was also observed with the increase in the proportion of food waste in the AcoD of food waste with SS [43]. According to the kinetic evaluation results, AcoD with SS did not provide any synergistic effect on the hydrolysis rate of the studied AWs, similar to methane production reported in Section 2.2. The above findings indicate that the AcoD with SS is not a suitable strategy for the enhancement of the methane production of the studied AWs. On the other hand, the AcoD of AWs and SS is an appropriate and practical approach for boosting methane production from SS, when waste management is an important concern.

## 3. Materials and Methods

### 3.1. Characterization of Agricultural Wastes and Sewage Sludge

Fresh AWs consisting of roots, stems, leaves and fruits of tomato, pepper, cucumber, eggplant and courgette were obtained from local greenhouses in Antalya, Turkey. The types and amounts of the wastes selected to prepare the mixed AWs sample were determined according to the local production statistics gathered from the Turkish Statistical Institute [44]. Mixed AWs consisting of 61.71% tomato, 22.44% cucumber, 7.92% eggplant, 5.72% green pepper and 2.21% courgette were ground to a 4–5 mm particle size and stored in sealed plastic bags at −20 °C until use for the composition analysis, pretreatment and BMP tests.

Sewage sludge (ss) consisting of 30% primary sludge and 70% waste-activated sludge was collected from the belt thickener of the Hurma Advanced Municipal Wastewater Treatment Plant (WWTP) in Antalya. Details of the WWTP were reported in the study of Perendeci et al. [45]. The SS sample was also stored at −20 °C until use for the composition analysis and BMP tests.

The prepared AWs sample was freeze-dried and ground to an average of a 1 mm particle size for the composition analysis. The analysis of total solids (TSs), organic matter (VS) and chemical oxygen demand (COD) was performed according to standard methods [46]. The content of lignin, cellulose, hemicellulose and soluble matter was determined by the FibreBag system (Gerhardt), using procedures proposed by van Soest [47]. Total Kjeldahl nitrogen (TKN) was determined using the total Kjeldahl nitrogen analyzer, as shown in Table 4. Carbohydrate concentration was determined as glucose, using the anthrone method based on quantifying the carbonyl functions (C=O) [48]. Protein concentration was determined according to the Lowry method [49]. Extractive matter including lipids was determined by accelerated solvent extraction (ASE) following the procedure by Bridoux et al. [50]. Elemental analysis was performed using the CHNS Elemental Analyzer (LECO, CHNS-932). The characterization of the raw AWs mixture is shown in Table 4 together with the characterization of SS.

### 3.2. Thermal and Thermochemical Pretreatment

Thermochemical pretreatments were applied on AWs with sulfuric acid (H_2_SO_4_), hydrochloric acid (HCl), sodium hydroxide (NaOH) or calcium hydroxide (Ca(OH)_2_). Thermal and thermochemical pretreatment experiments were carried out at the optimum conditions previously reported for the enhanced methane production [2,51,52,53]. The pretreatment conditions are summarized in Table 5.

Thermal and thermochemical pretreatment experiments were conducted in a 0.5 L laboratory-scale glass reactor immersed in an oil heating bath equipped with a reflux condenser. All pretreatment experiments were performed in duplicates. In the thermochemical pretreatment experiments, reactors were heated to predetermined temperatures after the addition of a chemical reagent. When the desired temperature was reached, 1 h of pretreatment time was initiated by introducing a predetermined amount of AWs. After the 1 h pretreatment time, the reactor was removed from the oil bath, and the reactor vessel was immersed in an ice and water bath. The content of the reactor cooled down to room temperature in 5–10 min. Total solids, VS and COD analyses were performed according to standard methods [46] before and after the BMP test. All analyses were carried out in duplicate. The remaining samples were stored in sealed plastic bags at −20 °C until the BMP tests.

### 3.3. Biochemical Methane Potential (BMP)

Batch BMP tests were carried out with the mixture of raw or pretreated AWs and SS at mixing ratios of 100:0, 80:20, 60:40, 50:50, 40:60, 20:80 and 0:100 based on the VS content of the wastes. Methane production was measured by the batch BMP tests under mesophilic (35 °C) conditions following the procedures established by Perendeci et al. [54]. Raw or pretreated AWs and SS were placed into a 500 mL BMP reactor with anaerobic seed sludge from a sugar beet factory digester of the anaerobic wastewater treatment plant (Eregli, Turkey). The substrate-to-inoculum (S/I) ratio was set as 0.5 (gVS waste/gVS anaerobic seed sludge) in the BMP reactors. Details of the BMP test are reported in the study of Gunerhan et al. [2].

The BMP tests lasted up to 100 days in the reactors incubated at 35 °C. Duplicate BMP tests were performed for all samples. Additionally, a BMP test was performed with three replicates with the inoculum (as control) to not take into account the methane produced by the anaerobic seed sludge. The amounts of methane produced from all samples and the inoculum were calculated as mL/gVS by dividing the calculated cumulative methane gas volumes by the amount of VS used at the start of the digestion tests and converting them to normal conditions of temperature and pressure. The results of the BMP tests were reported as the cumulative methane production (CMP). The CMP was calculated by subtracting the amounts of methane produced in inoculum (control) reactors from the amounts of methane produced in sample reactors.

The COD and VS removal efficiencies were calculated based on the measured concentrations in each reactor before and after the BMP tests. Synergistic and antagonistic effects were calculated according to the procedure described by Yilmaz et al. [37].

### 3.4. Digestion Kinetics

The effects of pretreatment and AcoD on anaerobic digestion kinetics were evaluated using the technical digestion time and first-order kinetics. The CMP obtained from raw AWs after 100 days of digestion time was assumed as the maximum methane production in the calculation of the technical digestion times (T_80_). Then, T_80_ values required to produce methane as much as 80% of the CMP obtained from raw AWs were calculated for each pretreatment by the linear regression of CMPs (Appendix A).

Hydrolysis of complex organic material has been considered as a rate-limiting step in anaerobic degradation. The analysis of CMPs using the first-order kinetic model provided valuable information about the hydrolysis kinetics [1,4,5,41]. In order to determine the hydrolysis rate (*k_h_*) using the first-order kinetic model, it has been proposed to evaluate the initial days of the process where rapid methane production is observed in anaerobic digestion [24]. Therefore, model calibration studies were performed by adjusting the hydrolysis rate (*k_h_*) until the first-order kinetic modeling results adequately fitted the CMPs obtained during the first six days of the BMP tests. The first-order kinetic model equation applied to evaluate anaerobic monodigestion is given in Equation (1):*M_P_* = *P_M_* (1 − exp[−*k_h_* × *t*])(1)
where *M_P_* is the cumulative methane production (mL/gVS) at a digestion time (*t*), *P_M_* is the maximum methane potential (mL/gVS), *k_h_* is the first-order rate constant (d^−1^) and *t* is the digestion time (d).

To evaluate the effect of AcoD on the hydrolysis rate of each cosubstrate, the first-order kinetic model was modified by adding a second term to the first-order kinetic equation due to the different biodegradability levels of the codigested wastes. as proposed previously [55]. The modified first-order kinetic model equation applied to determine the hydrolysis rates in AcoD is given in Equation (2):*M_P_* = (*P_M_*_1_ (1 − exp[−*k_h_*_1_ × *t*])) + (*P_M_*_2_ (1 − exp[−*k_h_*_2_ × *t*]))(2)
where the first part of the equation (with the subscript 1) denotes the kinetic parameters of the AWs used in AcoD, and the second part of the equation (with the subscript 2) denotes the kinetic parameters of the SS. The hydrolysis rates were calculated by minimizing the least-square difference between the observed and predicted CMPs in the beginning of the digestion period (first 6 days). The optimization process ended when the change in the residual was less than the specified tolerance set on 1 × 10^−9^. Model simulations were performed using AQUASIM 2.0 [56].

## 4. Conclusions

The objective of the present study was to determine a suitable strategy for the enhancement of methane production from AWs consisting of roots, stalks, leaves and fruits that originated from the production of tomato, pepper, cucumber, eggplant and courgette. In this study, two possible strategies, namely pretreatment and codigestion, were evaluated in terms of methane production, hydrolysis rate and technical digestion time. As a pretreatment method, thermal and thermochemical (with H_2_SO_4_, HCl, NaOH and Ca(OH)_2_) pretreatments were applied. After pretreatment, AcoD was applied on raw and pretreated AWs with SS. While the methane potentials were evaluated as a cumulative methane production, the hydrolysis rates were obtained from the first-order kinetic model.

Thermochemical pretreatment with NaOH provides the maximum enhancement in the anaerobic digestion of AWs, with a 56% increase in methane production, 81.90% increase in the hydrolysis rate and a 79.63% decrease in the technical digestion time compared to raw AWs. While thermochemical pretreatment with H_2_SO_4_ caused a decrease in the CMP, thermochemical pretreatment with HCl and Ca(OH)_2_ resulted in 29.97% and 37.08% more methane production compared to raw AWs, respectively. A comparison between thermochemical pretreatment with HCl and Ca(OH)_2_ showed that the maximum methane production was superior in thermochemical pretreatment with Ca(OH)_2_, while thermochemical pretreatment with HCl significantly improved the hydrolysis rate and technical digestion time. Although a higher hydrolysis rate was observed in thermochemical pretreatment with H_2_SO_4_, the cumulative methane production was determined as lower than that of raw AWs at the end of the digestion period. These results indicate that thermochemical pretreatment with H_2_SO_4_ influenced the hydrolysis of organic material, whereas methane production was disturbed by inhibitors such as HMF and furfural produced during the pretreatment.

Contrary to pretreatment, AcoD did not enhance the methane production potential of raw and pretreated AWs. The hydrolysis rates determined for AcoD support the conclusion that AcoD with SS is not suitable for improving methane production from the studied AWs. Nevertheless, the most appropriate mixing ratio was determined to be 80:20 on VS bases among the mixing ratios from the experimental results of the AcoD of the studied AWs with SS for obtaining a maximum possible methane production without any antagonistic effect. Furthermore, the selection of enhancement strategies depends on the substrate type and waste management regulations. Waste management is of great concern, especially for touristic places, such as Antalya, so the AcoD of SS with different kinds of AWs is an appropriate and practical approach for the enhancement of methane production from SS.

## Figures and Tables

**Figure 1 molecules-26-04175-f001:**
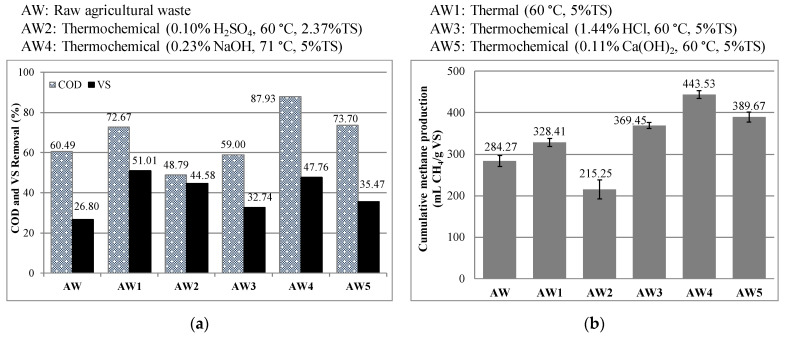
Effect of thermal and thermochemical pretreatment on (**a**) COD and VS removal and (**b**) CMP.

**Figure 2 molecules-26-04175-f002:**
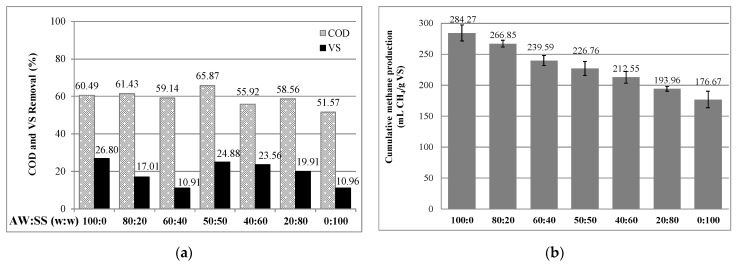
Effect of the AcoD of raw AWs with SS on (**a**) COD and VS removal, and (**b**) CMP.

**Figure 3 molecules-26-04175-f003:**
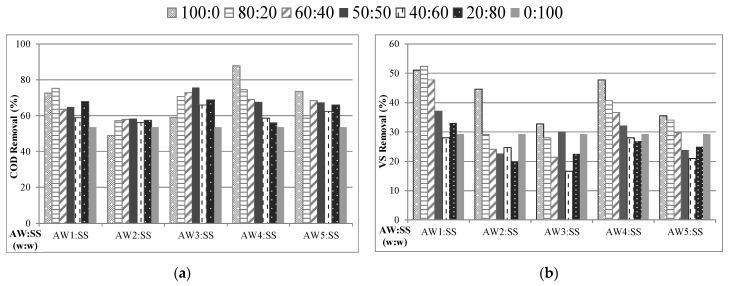
Effect of the AcoD of pretreated AW on (**a**) COD and (**b**) VS removal efficiencies.

**Figure 4 molecules-26-04175-f004:**
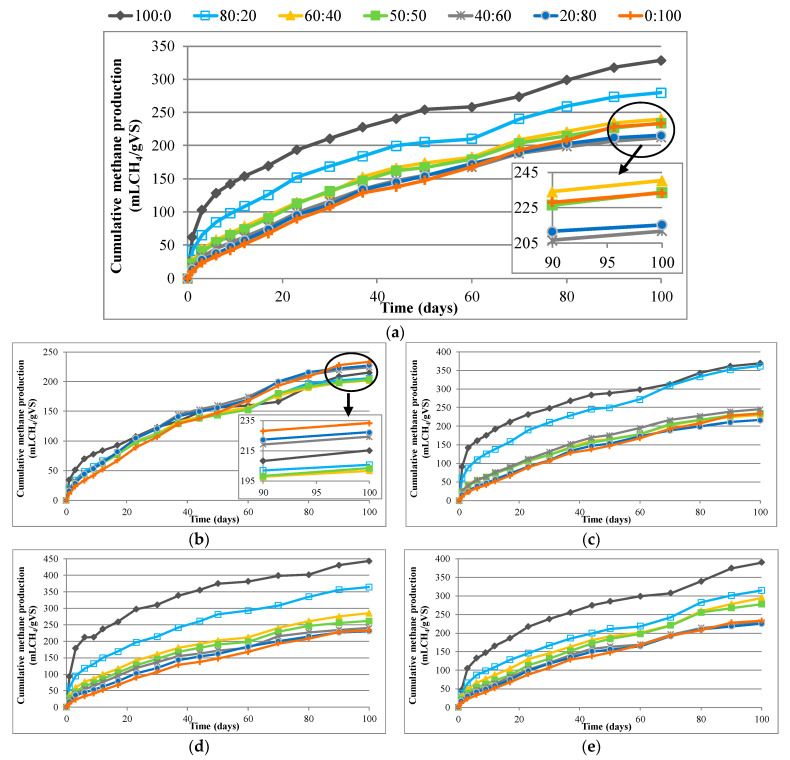
Average CMPs from the AcoD of (**a**) AW1, (**b**) AW2, (**c**) AW3, (**d**) AW4 and (**e**) AW5 with SS.

**Table 1 molecules-26-04175-t001:** Comparison of the measured and theoretical CMPs in the AcoD of raw AWs with SS.

Mixing Ratio (AWs/SS)	Measured CMP (mL/gVS)	Theoretical CMP (mL/gVS)	Synergistic and Antagonistic Constant (α)
100:0	284.269	-	
80:20	266.846	262.749	1.02
60:40	239.594	241.229	0.99
50:50	226.758	230.468	0.98
40:60	212.546	219.708	0.97
20:80	193.958	198.188	0.98
0:100	176.668	-	

**Table 2 molecules-26-04175-t002:** Kinetic parameters calculated for raw and pretreated AWs.

Sample	T_80_ (Days)	*k_h_* (1/Day)	*R* ^2^
AW	54	0.453	0.9964
AW1	37	0.604	0.9949
AW2	-	0.565	0.9785
AW3	22	0.816	0.9979
AW4	11	0.824	1.0000
AW5	27	0.462	0.9689

**Table 3 molecules-26-04175-t003:** Hydrolysis rates obtained for the AcoD of AWs and AW3 with SS.

AWs/SS (*w*/*w*)				AW3/SS (*w*/*w*)			
Mixing Ratio	*k_h_*_1_ (1/Day)	*k_h_*_2_ (1/Day)	*R* ^2^	Mixing Ratio	*k_h_*_1_ (1/Day)	*k_h_*_2_ (1/Day)	*R* ^2^
100:0	0.453	-	0.9964	100:0	0.816	-	0.9979
80:20	0.335	0.411	0.9955	80:20	0.625	0.305	0.9837
60:40	0.335	0.411	0.9958	60:40	0.625	0.305	0.9922
50:50	0.335	0.411	0.9929	50:50	0.625	0.305	0.9911
40:60	0.335	0.411	0.9984	40:60	0.625	0.305	0.9730
20:80	0.314	0.365	0.9885	20:80	0.348	0.305	0.9768
0:100	-	0.305	0.9952	0:100	-	0.305	0.9995

**Table 4 molecules-26-04175-t004:** Characterization of AWs and SS.

Parameter		AWs	SS
TS (gTS/kgSample)		913.93	165.01
VS (gVS/kgSample)		694.09	117.76
TKN (gTKN/kgVS)		24.80	467.26
Total COD (mgCOD/gVS)		1047	17948
Protein (g/kgVS)		94.56	1174.17
Extractable materials (g/kgVS)		42.17	0.58
Carbohydrate (g glucose/kgVS)		279.50	727.73
Soluble matter (%)		46.86	53.37
Hemicellulose (%)		14.45	22.27
Cellulose (%)		26.40	6.32
Lignin (%)		12.28	18.04
Elemental Composition (%)	Carbon (C)	34.16	43.43
	Hydrogen (H)	5.03	6.14
	Nitrogen (N)	2.39	6.84
	Sulfur (S)	0.82	0.71

**Table 5 molecules-26-04175-t005:** Pretreatment conditions of the AWs used for the AcoD with SS in this study.

Sample	Pretreatment Conditions	TS (%)	Temperature (°C)	Chemical Agent (%)	Reference
AW1	Thermal	5.0	60	-	[51]
AW2	Thermochemical (acidic agents)	2.37	60	0.10% H_2_SO_4_	[51]
AW3		5.0	60	1.44% HCl	[2]
AW4	Thermochemical (alkaline agents)	5.0	71	0.23% NaOH	[53]
AW5		5.0	60	0.11% Ca(OH)_2_	[52]

## Data Availability

Not available.

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
