# Peer review of "Pretreatment, Anaerobic Codigestion, or Both? Which Is More Suitable for the Enhancement of Methane Production from Agricultural Waste?"

_molecules, 2021, doi:10.3390/molecules26144175_

Round 1
Reviewer 1 Report
I have read the manuscript entitled "Pretreatment, anaerobic co-digestion or both? Which is more suitable for enhancement of methane production from agricultural waste" by L. Dumlu and co-authors. The report in interesting and deserves acceptance for publication after minor revision.
- L23: Percentage sign "%" is usually placed after numbers. See line 166 as well.
- L33-34: Delete "in terms of" and replace with "for". The phrase should read: "...types of wastes for energy production..."
- L130: Figure 1 should have a label for the x-axis.
- L172: delete "the".
- L207: Figure 2, provide axis label for x-axis. Provide the unit for the mixing ratio (e.g. w/w, v/v, etc). I observed that there was no unit for the mixing ratio throughout the manuscript, including materials and methods section. There needs to be clarity whether substrates were mixed as weight per weight or volume per volume, or otherwise.
- L254: Figure 3, provide units and axis label for x-axis.
- L268: Replace "given" with "presented".
- L299: Replace "framework" with "work" or "study".
- L325: Replace "obtained" with "recorded".
- L382: Table 3, provide unit or the mixing ratio.
- L508: Replace "is" with "was" because the work is now completed.
- L512: Replace "framework" with "work" or "study".
- A major concern with the discussion section is that it is limited to studies that considered related substrates. What is the performance of other substrates when considering AcoD. The authors should consider comparing with works that considered different substrates for AcoD. A paragraph in the Discussion Section could suffice. The should consider the following recent papers and revise the section accordingly: https://doi.org/10.1177/0734242X20975092 and https://doi.org/10.1186/s42834-020-00056-6.
Author Response
Answers to reviewer 1 comments can be found below the file.

Reviewer 2 Report
Good manuscript. Very accurate in writing and commenting on the results. The research on how to increase the methane produced by anaerobic digestion is fundamental as it is necessary to produce more biogas from alternative feedstocks that are not always of excellent quality. Thus, the authors have chosen a very interesting topic since the pretreatments and co-digestion of these alternative feedstocks are needed to enhance methane production.Minor amendments are required before the publication.
Introduction
P2 L45. Please, remove “However” and start the period with “The lignocellulosic nature…”.
P2 L79-80. Please, move the period “In addition to increase in biogas production due to the positive synergistic effect of different biomasses with complementary characteristics (especially in terms of the C/N ratio)” at line 83 after “[18-19].”
P2 L84. Please, replace “With” with “In the”.
P2 L86-87. Please, delete “Wastes that are anaerobically co-digested should be complementary, especially in terms of C/N ratio” because è stato già ditto a linea 79-81
P2 L87. Please, delete “Therefore”, start with “Sewage sludge”
Results and Discussion
P4 L150. Please, remove this comparison with Li et al. [16]. The comparison is not correct. The mixture used in the current study was made up of various residues, with a predominance of tomatoes; but there were other residues that could have a higher BMP and, taking into account their contribution in the mixture, can bring the BMP of AWs higher than just tomato residues. Authors can compare the raw AWs BMP to other feedstocks used in anaerobic digestion (i.e. maize silage, pig slurry, cattle mmanure, food waste).
P4 L160. Please, riporta % dopo i numeri. And replace “raw waste” with “untreated control”.
P4 L170. Please, riporta % dopo i numeri.
P2 L45. Please, remove the comma after “system” and put "actually" between two commas.
P5 L205. Please, remove “and 100%” and replace with “In addition, monodigestion of AWs and SS was carried out as control”.
P5 L215. Please, replace “done” with “performed”.
P6 L248. Please, remove “100%” and replace with “and compared with monodigestion of pretreated AWs and untreated SS”.
Materials and methods
P11 L420. The authors could identify here Total Kjeldahl nitrogen with TKN as it appears in the Table 4.
P11 L420. Please, replace “the” with “a”.
P11 L420. Please, add a comma after “glucose”.
P11 L424. Please, replace “based on the modification Bridoux et al. method” with “following Bridoux et al. [48] eith some modifications”. Please describe briefly the modifications.
P11 L433. Please, add a comma after “(NaOH) and replace “and” with “or”
P11 L436. Please, remove “which were validated to be optimal for thermochemical pretreatment of AWs in previous studies and applied in this study”.
P12 L446. Please, replace “to” with “-“. Write “Total solids” instead “TS” at the start of the phrase.
P12 L459. Please, replace “food” with “substrate” and “microorganisms” with “inoculum” (S/I).
P12 L463. “Times” are the replicates? If yes, please replace “times” with “replicates”.
Conclusion
Please remove “AW2, AW5, and AW3” between lines 521-522.
Author Response
Answers to reviewer 2 comments can be found below the file.

Reviewer 3 Report
In this manuscript, Perendeci et al. studied the effects of pretreatment (different thermochemical approaches) and co-digestion on methane production and hydrolysis rate in anaerobic digestion of agricultural wastes, and found the best strategy was the thermochemical pretreatment with sodium hydroxide. It was found 56% increase in methane production, 81.9% increase in hydrolysis rate and 79.6% decrease in technical digestion time compared to raw agricultural wastes. Such research is important for local environment. I recommend it to be published in Molecules. Below are my comments:
- In Introduction, the description of biochemical methane production (line 96-109) should be moved to Results and Discussion or Materials and Methods.
- In Figure 1, the detailed information of AW1 to 5 may be needed to be given in caption. It is difficult for reader to find what they are for.
- For the effect of AcoD of raw AWs with SS, it can be seen the CMP linearly decreased with increasing the content of SS, why the authors said the 80:20 is best ratio? What about 90:10 or 90:5?
- Lot of typos need to be revised, such put the % in front of number.
Author Response
Answers to reviewer 3 comments can be found below the file.
